# Impact Response of Aramid Fabric-Reinforced Polybenzoxazine/Urethane Composites Containing Multiwalled Carbon Nanotubes Used as Support Panel in Hard Armor

**DOI:** 10.3390/polym13162779

**Published:** 2021-08-18

**Authors:** Phattarin Mora, Chanchira Jubsilp, Christopher W. Bielawski, Sarawut Rimdusit

**Affiliations:** 1Research Unit in Polymeric Materials for Medical Practice Devices, Department of Chemical Engineering, Faculty of Engineering, Chulalongkorn University, Bangkok 10330, Thailand; Phattarin.m@gmail.com; 2Department of Chemical Engineering, Faculty of Engineering, Srinakharinwirot University, Nakhonnayok 26120, Thailand; chanchira@g.swu.ac.th; 3Center for Multidimensional Carbon Materials (CMCM), Institute for Basic Science (IBS), Ulsan 44919, Korea; bielawski@unist.ac.kr; 4Department of Chemistry, Ulsan National Institute of Science and Technology (UNIST), Ulsan 44919, Korea

**Keywords:** polybenzoxazine composite, multiwalled carbon nanotube, nanocomposite, energy-absorption, numerical simulation

## Abstract

The aim of this research project is to analyze support panels that are based on aramid fabrics which are reinforced with polybenzoxazine/urethane (poly(BA-a/PU)) composites and contain multiwalled carbon nanotubes (MWCNTs). Through the measurement of mechanical properties and a series of ballistic-impact tests that used 7.62 × 51 mm^2^ projectiles (National Institute of Justice (NIJ), level III), the incorporated MWCNTs were found to enhance the energy-absorption (*E_Abs_*) property of the composites, improve ballistic performance, and reduce damage. The perforation process and the ballistic limit (*V*_50_) of the composite were also studied via numerical simulation, and the calculated damage patterns were correlated with the experimental results. The result indicated hard armor based on polybenzoxazine nanocomposites could completely protect the perforation of a 7.62 × 51 mm^2^ projectile at impact velocity range of 847 ± 9.1 m/s. The results revealed the potential for using the poly(BA-a/PU) nanocomposites as energy-absorption panels for hard armor.

## 1. Introduction

Fiber-reinforced polymer composites (FRPs) have been extensively used in many high-performance applications that pertain to ballistic protection, such as helmets [1] and other types of body armor [2,3,4,5,6]. The utility is due to the excellent mechanical properties and energy-absorption capabilities of FRPs. Among the fibers used in such applications, aramid woven fabrics are often effective and commonly used in back-energy-absorption panels. When such panels are juxtaposed with front ceramic tiles, they are often referred to as “ballistic hard armor” in part because they mitigate high-velocity projectiles, including those that adhere to National Institute of Justice (NIJ) standards (e.g., 7.62 × 51 mm^2^ caliber projectiles with an average impact velocity of 847 ± 9.1 m/s) [7,8,9,10]. A primary role of the energy-absorption panel is to dissipate and absorb the kinetic energy of projectile fragments as they are destroyed by the front ceramic tiles. As such, the energy-absorption properties of the composite are of paramount importance for evaluating the ballistic performance of the support panel.

Many factors affect the energy-absorption properties exhibited by impacted FRP composites, including the type of fiber [11,12] and polymer matrix used [13], as optimal results require a suitable interaction between these two components [13,14]. Polybenzoxazines are a relatively new class of phenolic resin that are finding utility in an increasing number of applications [15]. The material is prepared by subjecting a cyclic benzoxazine monomer to a ring-opening polymerization reaction followed heat treatment to induce curing. The polymerization and curing steps do not require a catalyst and proceed without the release of by-products. Since the resulting resin features phenol function groups, the material can be chemically modified [16,17] or used to enhance interactions with fibers or fillers [18,19]. For example, hybrids of polybenzoxazines with other polymeric materials, including polyurethanes (PU) [20,21,22], novolac-type resins [23,24], epoxy resins [22,25,26], and dianhydride resins [25,27,28] have been reported. Takeichi et al. [26] examined a benzoxazine–urethane copolymer in detail, and concluded that crosslinking between the two monomers occurs during polymerization and leads to the formation of polymer networks. Okhawilai et al. [18] subsequently determined that the interactions formed between poly(benzoxazine-co-urethane) matrices and aramids are enhanced when the polyurethane content of the composite is in the range of 0–40 wt%. For example, a poly(benzoxazine–co-urethane) composite with 20 wt% of urethane mixed with an aramid fiber was subjected to 9 mm projectiles (NIJ standard #0101.04, level II and IIIA) to assess the tensile and energy-absorption characteristics of the composite. Based on the results, the composite was deemed suitable for use in soft armor applications. The ballistic-impact response of the composite was not tested against a more powerful 7.62 × 51 mm^2^ projectile (NIJ level III). Based on these observations, it was reasoned that aramid fabrics that were reinforced with a polybenzoxazine/urethane composite may be suitable for use in such applications.

Nanoparticles have been used as additives to significantly improve the performance displayed by FRP composites, such as their mechanical properties, stiffness, and energy-absorption capability [29]. Gibson et al. [30] studied the ballistic performance of a plain-weave Kevlar^®^29 reinforced epoxy that contained carbon nanotubes or milled fibers for use in hard ballistic armor applications. Improvements in ballistic limit velocity were realized upon the addition of the carbon nanotubes (0.5 wt%) as well as the milled fiber (1.65 wt%). Pandya et al. [31] found that the addition of carbon nanotubes to glass fiber-reinforced epoxy composites resulted in an enhanced ballistic limit velocity (*V*_50_) as well as a higher energy-absorption capability. Collectively, these and other discoveries indicate that nanomaterials, especially carbon nanotubes, may be used to enhance the performance of FRP composites with a minimal weight penalty and while retaining a relatively low cost.

Since the ballistic response and perforation process is complex, numerical simulations have been used to better understand the deformation and damage areas that are created upon impact [20,32,33]. Zhang et al. [34] studied the effect of pre-stress on the ballistic performance displayed by bi-layer ceramic composites in armor applications using a combination of experiments and calculations. A numerical technique was used to evaluate the penetration process of a projectile as well as the mechanism of pre-stress, and the effectiveness of the simulation results was separately validated by experiment. KılıÇ et al. [35] investigated the performance of steel armor plates impacted by 7.62 mm armor piercing (AP) rounds using a combination of finite element methods (FEM) and an artificial neural network (ANN). The former was also used to predict ballistic limit thickness and the depth of penetration. The penetration process and the deformation within the impacted zone were also simulated.

Based on this background information, we reasoned that aramid fabrics reinforced with a poly(BA-a/PU) composite that contains multiwalled carbon nanotubes (MWCNTs) may offer potential for use as an energy-absorption panels, particularly those that are impacted by rigid 7.62 × 51 mm^2^ projectiles. The added MWCNT was further expected to modulate the mechanical properties displayed by the composite, including strength, modulus, and ballistic performance. Likewise, the interfacial interaction between the poly(BA-a/PU) matrix, the MWCNT, and the aramid fabric was also expected to be a factor. As described below, these points were investigated through a range of spectroscopy, mechanical, and ballistic tests. Furthermore, a numerical analysis of the ballistic behavior exhibited by the composites was simulated using a S-glass fabric-reinforced poly(BA-a) composite as a strike panel and an aramid fabric that was reinforced with a poly(BA-a)/PU composite containing MWCNTs as a support panel, and the data obtained from the analysis were compared to experimental results.

## 2. Materials and Methods

### 2.1. Materials

A benzoxazine-based monomer, bis(3-phenyl-3,4-dihydro-2H-1,3-benzoxinyl) isopropane (BA-a), was prepared in situ by mixing 2,2′-bis(4-hydroxyphenyl)-propane (bisphenol-A), aniline, and formaldehyde. Bisphenol-A (polycarbonate grade) was kindly supplied by PTT Phenol Co., Ltd. (Rayong, Thailand). Formaldehyde (AR grade) was purchased from Merck Co., Ltd. (Darmstadt, Germany) Aniline (AR grade) was purchased from Panreac Quimica, S.A. (Barcelona, Spain). A urethane prepolymer (PU) was synthesized from 2,4-toluene diisocyanate (TDI) and propylene glycol (PPG) with a number average molecular weight (M_n_) of 2000 g/mole; these materials were provided by IRPC Public Company, Ltd., (Rayong, Thailand). A plain weave of an aramid fabric with an area weight density of 340 g/m^2^ was purchased from the Thai Polyadd Limited Partnership (Bangkok, Thailand). Multiwalled carbon nanotubes (MWCNTs) were purchased from Nano Generation Co., Ltd. (Chiang Mai, Thailand). The MWCNTs featured an outer diameter of 12.9 nm and a tube length of 3–12 μm. The density of the MWCNTs was 1.90 g/cm^3^. All chemicals were used as received.

### 2.2. Resin Preparation

The benzoxazine monomer was synthesized from bisphenol-A, formaldehyde, and aniline at a molar ratio of 1:4:2 using the solvent-less synthesis technique [36]. The three reactants were continuously stirred at 110 °C for approximately 40 min. The resulting monomer was a clear, yellowish color and a solid at room temperature. The solid monomer was ground into fine powder before being used in copolymer formulations. The urethane prepolymer was prepared from TDI and PPG at a molar ratio of 2:1. The two reactants were mixed in a four-necked round-bottomed flask under a stream of nitrogen gas at a temperature of 70 °C for 2 h to yield a light yellow prepolymer. After cooling to room temperature, the urethane prepolymer was kept in a closed container that was with purged nitrogen gas and then stored in a refrigerator.

### 2.3. Sample Preparation

The nanocomposite was prepared from an aramid fabric that was reinforced with a poly(BA-a/PU) composite containing MWCNTs. The polybenzoxazine (BA-a) was combined with the urethane prepolymer to yield a BA-a/PU mixture containing 20 wt% of the polyurethane (PU) prepolymer. The MWCNTs were dried at 80 °C for 24 h in an air-circulated oven (Worldco Co, Ltd., Bangkok, Thailand) until a constant weight was achieved and then kept in a desiccator at room temperature. Afterward, various quantities (i.e., 0.0, 0.1, 0.3, 0.5, or 1.0 wt%) of the MWCNTs were added to PU prepolymer at room temperature for 24 h to obtain a well dispersed mold following by gentle mixing with BA-a resin at 120 °C for 40 min to achieve a dispersed compound. The compound was coated onto an aramid fabric using the hand-layup technique at 120 °C to afford the prepregs, which was then heated to a temperature of 160 °C for 40 min, 180 °C for 20 min, 200 °C for 10 min and then cured at a temperature of 200 °C for 2 h in a compression molder (model LP20-B from Labtech engineering Co., Ltd., Bangkok, Thailand) at a pressure of 10 MPa. The weight fraction of the fiber was kept constant at 75 wt%. All samples were air-cooled to room temperature after opening the mold and then cut with a diamond blade into shapes required for characterization.

### 2.4. Sample Characterization

#### 2.4.1. Fourier Transform Infrared Spectroscopy (FT-IR)

Fourier transform infrared spectra were carried out on a Perkin Elmer Spectrum GX FT-IR (Perkin Elmer Co., Ltd., Waltham, MA, USA) spectrometer equipped with an ATR accessory. All spectra were taken with 64 scans at a resolution of 4 cm^−1^ over a spectral range of 4000–650 cm^−1^.

#### 2.4.2. Mechanical Tests

The tensile tests of an 8-ply aramid fabric that was reinforced with a BA-a/PU composite containing MWCNTs (mass content in the range of 0.0–1.0 wt%) were conducted according to the ASTM D3039 standard. The specimen dimension was 130 × 25 × 2.97 mm^3^. A 10 mm diameter tensile specimen featured a 70 mm gauge length. The length of the grip section was 27.5 mm. The test specimens were performed with a crosshead speed of 2 mm/min using a Universal Testing Machine model 8872, Instron Co., Ltd., (Bangkok, Thailand). A minimum of eight samples were tested and averaged values were determined.

#### 2.4.3. Morphological Assessment

Interfacial bonding was evaluated using a scanning electron microscope (SEM, model JSM-6510A from JEOL Ltd. (Tokyo, Japan)) operating at an acceleration voltage of 20 kV. All samples were coated with a thin layer of gold film using a sputter coater (model SCD 040 from Oerlikon Balzers Coating Co. Ltd., Chonburi, Thailand) to render the surfaces conductive.

#### 2.4.4. Ballistic Tests

The ballistic-impact tests were conducted using cartridges that propel projectiles onto the composite specimens. Two chronograph units were placed in front of and behind the target holding unit to record the impact and residual velocities of the projectiles, respectively. The experimental setup is shown in Figure 1. The 150 × 150 mm^2^ composite plate was clamped to the target holder. Following the Level III NIJ standard, the tests were conducted using 7.62 × 51 mm^2^ projectiles at an impact velocity of 847 ± 9.1 m/s and at an angle of 90° to the specimen with one shot at the center. The projectiles were measured to have a diameter of 7.79 mm and a mass of 9.65 g. The equation used to measure the energy absorption of the samples during an impact is shown in Equation (1) [5].
*E_Abs_* = 1/2*m_p_* (*V_s_*^2^ − *V_r_*^2^)(1)
where *m_p_* is the mass of the projectile (kg), *V**_s_* is the impact projectile velocity (m/s) and *V_r_* is the residual projectile velocity after penetrating through the sample (m/s). The ballistic limit velocity (*V_bl_*) was calculated according to Equation (2)
*V_bl_* = (*V_s_*^2^ − *V_r_*^2^)^1/2^(2)

#### 2.4.5. Numerical Simulations

The ballistic performance and the damage pattern of an aramid fabric-reinforced poly(BA-a/PU) composite containing MWCNT was studied using a series of numerical simulations. The impact test on specimens with dimensions of 150 × 150 mm^2^ was tested using 7.62 × 51 mm^2^ projectiles at various impact velocities. The perforation process and deformation of the specimen was evaluated numerically with a commercial ANSYS AUTODYN system (ANSYS, Inc., (Canonsburg, PA, USA)). The specimen was simulated as an orthotropic material in which the material properties differ along the three orthogonal planes. The orthotropic equation of state (EOS) function allows a nonlinear fit when coupled with an orthotropic stiffness matrix. The fabrics were assumed to have identical properties in all directions but dependent on thickness. It was also assumed that the materials were homogeneous. The ballistic panel model was created with four fixed edges. A 7.62 × 51 mm^2^ projectile made of a brass jacket with a lead core was modeled following the strength models of Steinberg-Guinan. Table 1 summarizes the material properties of the projectile obtained from the standard ANSYS AUTODYN material library [37]. The geometry of the projectile is shown in Figure 2. To facilitate comparison, the simulation was performed under the same conditions as used in the experimental procedures. To validate the input properties of the materials, the deformation of the panel observed from the experimental and numerical results was systematically compared as this procedure was widely used for material property validation. The deformation patterns of the composite panels with dimensions of 150 × 150 mm^2^ were deduced from the simulated ballistic impacts. The *V*_50_ of the composite system, defined as the incident impact velocity at which there is a 50% probability of partial penetration and a 50% probability of perforation, was numerically estimated from the ballistic panel. The ballistic limit was determined by varying impact velocities of the projectile until the sample was perforated. The determined *V*_50_ value was the average of the highest partial penetration velocities and the lowest complete penetration velocities at an equal impact number of three. The estimated energy absorption of the hard armor composites was calculated according to Equation (3).
*E_Abs_* = 1/2*m_p_V_50_*^2^(3)
where *E_Abs_* is the energy absorption of the nanocomposites (J), *m_p_* is the mass of the projectile (kg) = 0.00965 kg [32] and *V*_50_ is the ballistic limit velocity (m/s).

## 3. Results

### 3.1. Synthesis of the Benzoxazine/Urethane Copolymers Containing MWCNTs

The chemical structure of the BA-a monomer and its corresponding polymer, poly(BA-a), was characterized using FT-IR spectroscopy. As shown in Figure 3a, the BA-a monomer featured an absorption peak at 1230 cm^−1^ which was assigned to the aromatic ether (C-O-C stretching) of its oxazine unit. The bands at 936 cm^−1^ and 1488 cm^−1^ were attributed to the trisubstituted aryl moiety [26]. After thermal curing, these bands disappeared, which indicated that the oxazine reacted and underwent ring-opening. In addition, new absorption peaks at 878 cm^−1^ and 1477 cm^−1^, consistent with a tetra-substituted arene, were observed, consistent with the ring-opening reaction as shown in Figure 3b [26,33]. A further indication of the ring-opening reaction was evident by the appearance of a broad signal at about 3380 cm^−1^ which was assigned to a phenol, a functional group that can react with the isocyanate group (NCO) of the urethane prepolymer [38,39].

The formation of networks between the BA-a monomer and the urethane prepolymer, both in virgin (unfilled) and filled (0.1 wt% MWCNT) forms, after thermal curing was also investigated by FT-IR spectroscopy; key results are shown in Figure 3c. Absorption signals observed at 1230 cm^−1^ (C–O–C stretching mode of benzoxazine ring), 1488 cm^−1^ and 936 cm^−1^ (trisubstituted arene ring of the BA-a monomer) and 2280 cm^−1^ (NCO group of urethane prepolymer) were absent from the spectrum [26,33]. Meanwhile, new absorbances appeared at 1730 cm^−1^ and 1490 cm^−1^, and were assigned to the urethane carbonyl (C=O) and a secondary urethane amide (C–NH), respectively. Collectively, these results indicated that the polybenzoxazine underwent ring-opening and afforded a benzoxazine–urethane as a product [26]. Moreover, the appearance of a broad signal at about 3300 cm^−1^ was observed and assigned to the phenolic hydroxyl group of the polybenzoxazine. In the FT-IR spectrum recorded for a pristine MWCNT (Figure 3d), a signal at 1540 cm^−1^ was observed and attributed to the presence of C=C bonds. An FT-IR spectrum recorded for poly(BA-a/PU) composite containing MWCNTs is shown in Figure 3e and found to be generally similar to that of poly(BA-a/PU). Signals assigned to the C–O–C stretching frequency of the benzoxazine ring (1230 cm^−1^), the trisubstituted arene ring (1488 cm^−1^ and 936 cm^−1^) and the NCO group of urethane prepolymer (2280 cm^−1^) were absent whereas signals attributed to the urethane carbonyl (1730 cm^−1^) and a secondary urethane amide (1490 cm^−1^) were observed [26,33]. The appearance of a broad signal at 3300 cm^−1^, which was assigned to the phenolic hydroxyl group of polybenzoxazine, was identified and may stem from interfacial contact with the aramid fabric [18,19]. From these results, we concluded that the incorporation of MWCNT may not hinder interaction between poly(BA-a/PU) matrix and aramid fabric, and that the interaction between the poly(BA-a/PU) matrix, the MWCNT nanoparticles, and aramid fabric could be promoted by the hydroxyl groups present in the polymer matrix. A possible chemical reaction between the BA-a monomer and the PU prepolymer is shown in Figure 4. The BA-a oxazine may undergo ring-opening at a suitable temperature, and the phenolic hydroxyl groups of the resulting product may then react with the isocyanate group of urethane prepolymer to form urethane carbonyl linkages in the copolymer networks. These results are also in good agreement with the previous reports by our group [38,39] and Takeichi et al. [26] using different types of resins.

### 3.2. Assessment of the Mechanical Properties of Aramid Fabrics That Were Reinforced with Poly(BA-a/PU) Composites Containing MWCNTs

Beyond the use of fiber-reinforced polymers (FRP) in ballistic protection, the demand for composites possessing excellent mechanical properties is increasing [40]. The effect of the added MWCNT on the tensile properties of the aramid fabric that was reinforced with a poly(BA-a/PU) composite was investigated; key results are shown in Figure 5. The tensile strength of the reinforced aramid fabric (no added MWCNT) was measured to be 454 MPa. For comparison, the tensile strength values increased to 515 MPa upon the addition of the 0.3 wt% MWCNT to the composite. The value measured for a reinforced aramid fabric containing more than 0.3 wt% of added MWCNTs was slightly decreased to 462 MPa. The enhancement in tensile strength may be due to a uniform dispersion and strong interfacial bonding between the added MWCNTs and the polymer matrix [41], while the slight decrease in tensile strength observed when more than 0.3 wt% of MWCNTs were added may be due to a poor dispersion and an increased aggregate formation of the MWCNT in the poly(BA-a/PU) matrix. The tensile modulus values of a reinforced aramid fabric containing MWCNTs were observed to increase from 24.5 to 26.8 GPa as the quantity of added MWCNT increased. Moreover, the measured values were higher than those determined for a reinforced aramid fabric that lacked the added MWCNT (23.2 GPa). The difference may be due to the MWCNT which adds rigidity to the poly(BA-a/PU) composites and contributes to the stiffness of the resulting nanocomposites [42]. The enhancement in the tensile modulus has the potential to reduce damaged area when used in ballistic applications [14].

### 3.3. Morphology of Aramid Fabric Reinforced Poly(BA-a/PU) Composites Filled with MWCNTs

Scanning electron microscopy (SEM) was used to visualize the dispersion of the MWCNTs in the poly(BA-a/PU) matrix. Figure 6 shows micrographs recorded for neat MWCNTs and pristine aramid fabric as well as a series of poly(BA-a/PU) matrices with various MWCNT loadings (e.g., 0, 0.1, 0.3, 0.5 and 1.0 wt%). As shown in Figure 6a, an image recorded for a neat aramid fabric (no added MWCNT) was found to exhibit spaces between the fibers. Inspection of aramid fabrics reinforced with poly(BA-a/PU) and containing 0.3 wt% of MWCNTs (Figure 6b) revealed that the surface was smooth, and that the aramid fabric was coated by the polymer matrix. However, as the MWCNT loading increased to 0.5 wt% (Figure 6c), the surface of corresponding nanocomposite found to be rough and the fabric was not fully coated with polymer in part because the MWCNTs underwent aggregation due to decreased matrix-fiber interactions [43]. Furthermore, the neat MWCNTs appeared to undergo agglomeration; see Figure 6d. Relatively good interfacial adhesion between these components was observed as the MWCNT was added to the poly(BA-a/PU) and the MWCNTs appeared to be well dispersed; for example, see Figure 6e which reflects a composite that contains 0.3 wt% of MWCNTs. As the quantity of added MWCNT increased to 0.5 wt%, air gaps and aggregation were detected on the fracture surface of the nanocomposites (see Figure 6f). As will be described below, composites containing 0.3 wt% MWCNT exhibited the highest strength and energy-absorption characteristics which, based on the SEM data, may be attributed to good interfacial adhesion between the polymer matrix, filler, and fiber. Likewise, use of higher or lower MWCNT loadings resulted in decreased mechanical performance due to decreased interfacial interactions.

### 3.4. Ballistic Assessment of Aramid Fabrics That Are Reinforced with Poly(BA-a/PU) Composites and Contain MWCNTs

An 8-ply aramid fabric that was reinforced with a poly(BA-a/PU) composite containing 0.3 wt% of MWCNTs was subjected to 7.62 × 51 mm^2^ projectiles at an impact velocity (*V_s_*) of 847 ± 9.1 m/s (NIJ Level III standard). The vs. and residual velocity (*V_r_*) values of the projectiles after the impact event, which were recorded using a chronograph, are summarized in Table 2. The corresponding energy absorption (*E_Abs_*) and ballistic limit velocity (*V_bl_*) values were calculated using Equations (1) and (2), respectively. The residual velocity was determined to be lower than the impact velocity for each specimen measured which may reflect the transfer of the kinetic energy of the projectile to the nanocomposites during the impact event. The energy-absorption value of the nanocomposite during the ballistic-impact test was 190.2 J, which was higher than the 143.4 J value measured for a poly(BA-a/PU) composite that lacked the MWCNT additive. Collectively, these results indicated that stress transfer between plies was effective and that the added MWCNTs enhanced energy absorption by facilitating the interfacial interactions between the polymer matrix, filler, and fiber. The energy absorption measured for the aramid fabric that was reinforced with poly(BA-a/PU) and contained 0.3 wt% of MWCNTs was higher than that of the composite that lacked the additive by up to 33%. Similarly, the ballistic limit velocity measured for the nanocomposite that contained 0.3 wt% of MWCNTs was 16% greater than that of the poly(BA-a/PU) composite that lacked the additive. These results revealed that adding MWCNTs to reinforced aramid fabric composites improved impact performance which may be due to an enhanced interfacial interaction between the polymer matrix, filler, and fiber.

### 3.5. Ballistic Response of Aramid Fabric Reinforced Poly(BA-a/PU) Composites Filled with MWCNT Using as Support Panel

Next, a series of impact tests were performed following the NIJ Level III standard. A S-glass fabric-reinforced poly(BA-a) composite was used as the strike panel and supported by an aramid fabric that was reinforced with the poly(BA-a/PU) composite containing 0.3 wt% of MWCNTs. An area of 150 × 150 mm^2^ was identified and subjected to a 7.62 × 51 mm^2^ projectile at an impact velocity of 847 ± 9.1 m/s. For comparison, the test was repeated using a support plate that lacked the MWCNT additive. Key results are summarized in Figure 7. The specimens that contained the MWCNT additive appeared to stop and catch the projectile as the strike panel was fully penetrated, but the support panel was only partially penetrated. In contrast, the specimens that lacked the MWCNT additive did not catch the projectile. The different outcomes may be explained by considering the impact process. Upon impacting the strike panel (i.e., S-glass fabric-reinforced poly(BA-a) composite), the projectile is broken up into smaller fragments which are then captured by the support panel. The support panel containing the MWCNT additive can do this more effectively because it exhibits a relatively high energy-absorption capability. Regardless, for both types of specimens, the strike panels exhibited fiber breakage, matrix cracking, and matrix delamination. Likewise, the support panels showed fiber breakage, tensile failure in the primary and secondary yarns, and cone formation. The diameter of the cone formation area on the backside of the support panel that contained the MWCNT additive was measured to be 73 mm, lower than the 100 mm value measured for the support panel that lacked the MWCNT additive. Moreover, as can be seen from a side view, the depth of the cone formation displayed by the specimen containing the MWCNT was 42% lower than the specimen which lacked the additive (i.e., 22 vs. 38 mm). Key results of impact response of support panel based on an aramid fabric that was reinforced with the poly(BA-a/PU) composite containing 0.3 wt% of MWCNTs are also summarized in Figure 8. Collectively, these observations further supported the conclusion that the energy-absorption characteristic specimen that contained the MWCNTs was relatively high.

### 3.6. Numerical Simulation Analyses of Multilayered Armor Based on Polybenzoxazine Nanocomposites

Numerical simulations were utilized to gain a deeper understanding of the projectile penetration process and to probe the underlying deformation mechanisms. The temporal evolution of the simulations are shown in Figure 9 and key material properties are presented in Table 3 [44]. Approximately 13 μs after the strike panel undergoes fracture, the support panel begins to fail with significant damage being evident at 32 μs after impact. The projectile appears to stop moving 189 μs after impact. Collectively, these results indicate that the propagation of the stress wave is fast even after the projectile reaches the support panel and that the polybenzoxazine-based composite containing the MWCNT can protect against 7.62 × 51 mm^2^ projectiles at a velocity of up to 848 m/s.

### 3.7. Comparison of the Experimental and Numerical Results

The experimental and numerical results of the ballistic-impact tests were next compared. As can be observed in Figure 10, which presents the damaged front and rear sides of the impacted specimens, the simulation qualitatively predicted the extent of the damage. The damage area of the support panel was measured to be 21 mm × 31.4 mm, while the calculated damage area was measured at 19.8 mm × 33 mm, a difference of approximately 1%. Likewise, the diameter of the circular perforation area was measured to be 73 mm, a result that favorably compares with the calculated value of 73.8 mm. The depth of penetration and the residual projectile shape after impact were also simulated and compared to experiment. As shown in Figure 11, the experimental results were in good agreement with the numerical simulations. The experimental depth of penetration extent was measured to be 22 mm whereas the numerical model predicted a depth of 21 mm, a difference of approximately 4.5%. Likewise, the diameter of the circular shape of the residual projectile was measured to be 10.0 mm, a value similar to that predicted by the simulation. Collectively, these results indicated that the experimental results were qualitatively and quantitatively in good agreement with the results predicted by the numerical simulation.

### 3.8. Ballistic Performance Predicted by Numerical Simulation of Multilayered Armor Based on Polybenzoxazine Nanocomposites

Numerical simulations were also used to evaluate key ballistic performance metrics, including the ballistic limit velocity (*V*_50_) and the energy absorption (*E_Abs_*), of multilayered hard armor specimens that consisted of a strike panel of a S-glass fabric composite and a support panel of an aramid fabric that was reinforced with a poly(BA-a/PU) composite containing MWCNTs. The specimens were subjected to a 7.62 × 51 mm^2^ projectile and then the *V*_50_ was evaluated by systematically varying the impact velocity in conjunction with the impact number. The average velocities required to achieve partial versus full perforation were determined and key results are shown in Figure 12. The residual velocity was estimated to be 0 m/s for partial penetration whereas, it was found to be increased with increasing impact velocity for full penetration. Based on these data, the ballistic limit velocity of the multilayered hard armor was calculated to be 925 m/s. A similar result was reported by Vasundhra et al. [45], who studied how plate thickness affects the *V*_50_ in Rolled Homogenous Armor (RHA) steel using 7.62 mm armor piercing projectiles with 854 m/s. The areal weight density of the RHA plate was reported to be 14.1 g/cm^2^ whereas the aramid fabric-reinforced poly(BA-a/PU) composite filled with MWCNT was determined to feature an areal weight density of 4.0 g/cm^2^. Using Equation (3), the estimated energy absorption (*E_Abs_*) for the polybenzoxazine composite was calculated to be as high as 4128 J. Moreover, the ballistic performance of our developed armor was compared to those of other systems tested with rifle projectile level for hard armor as summarized in Table 4. Collectively, these results indicated that multilayered hard armor specimens that consisted of a strike panel of S-glass fabric composite and supported by a panel of aramid fabric that was reinforced with a poly(BA-a/PU) composite containing MWCNTs could protect against the perforation of a 7.62 × 51 mm^2^ projectile at an impact velocity of up to 847 ± 9.1 m/s.

## 4. Conclusions

The effect of adding MWCNTs to aramid fabrics that are reinforced with a poly(BA-a/PU) composite was explored. Emphasis was placed on assessing how the mechanical properties and ballistic-impact performance of the composites were affected by the MWCNT additive. Adding MWCNTs to the composite significantly improved the tensile strength as well as the modulus of the material. The added MWCNT also improved the energy absorption and ballistic limit velocity by up to 33% and 16% respectively, when compared analogues that lacked the MWCNT additive. The improvement in energy absorption stems from interfacial adhesion between the aramid fabric, poly(BA-a/PU) matrix and added MWCNTs. Numerical simulations were also used to study the perforation process and to assess an ability to predict damage. Good agreement between the simulations and experimental results was realized, which reflected the accuracy of the underlying model. The results show that aramid fabrics that are reinforced with poly(BA-a/PU) composites which contain MWCNTs hold potential for use as energy-absorption panels in hard armor applications.

## Figures and Tables

**Figure 1 polymers-13-02779-f001:**
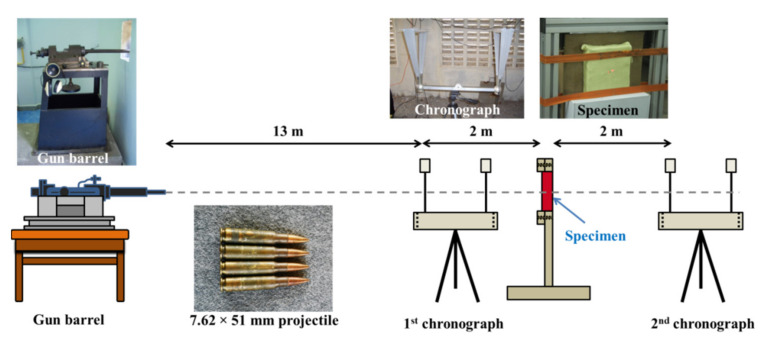
Illustration of the experimental setup used for the high-velocity impact tests.

**Figure 2 polymers-13-02779-f002:**
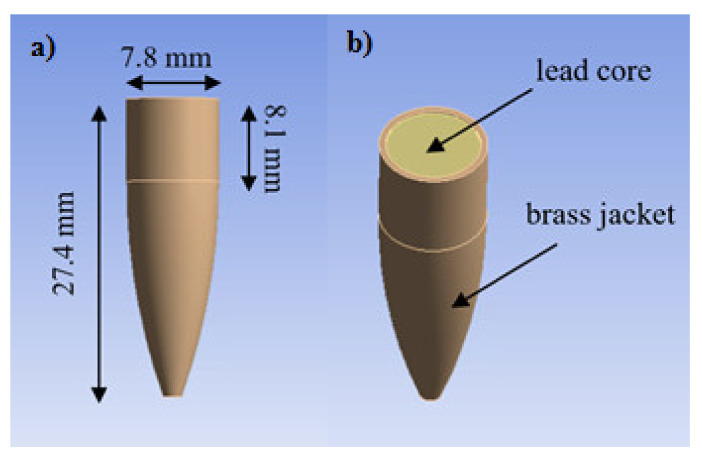
(**a**) The geometry of the 7.62 × 51 mm^2^ projectile and (**b**) the materials used in the jacket and core of the projectile.

**Figure 3 polymers-13-02779-f003:**
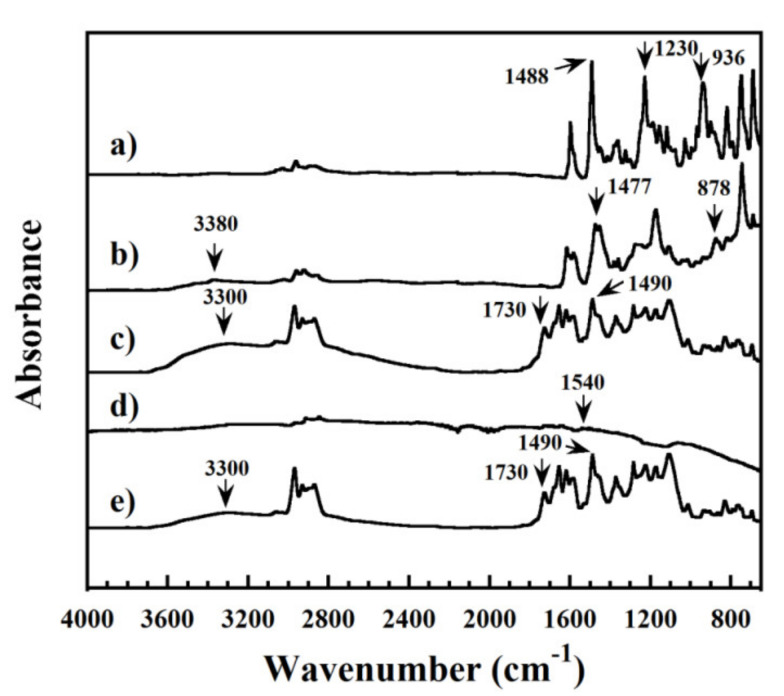
FT-IR spectra recorded for the (**a**) BA-a monomer, (**b**) poly(BA-a), (**c**) poly(BA-a/PU), (**d**) neat MWCNTs, and (**e**) poly(BA-a/PU) composite containing MWCNTs.

**Figure 4 polymers-13-02779-f004:**
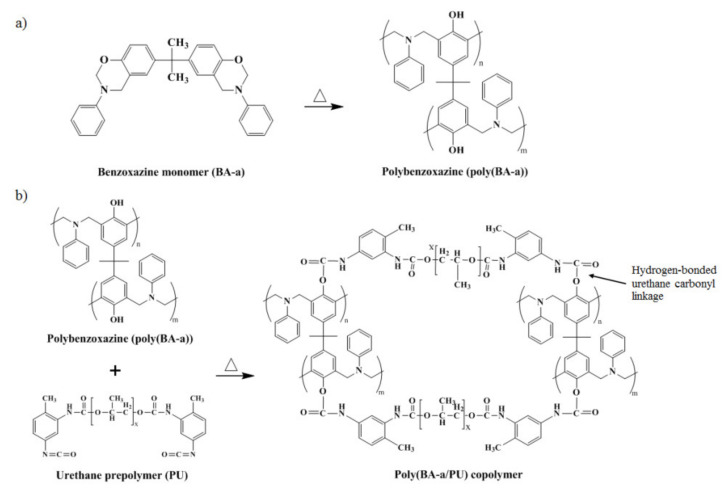
(**a**) Proposed curing reaction of the BA-a monomer and (**b**) a plausible chemical reaction between the poly(BA-a) and a PU prepolymer.

**Figure 5 polymers-13-02779-f005:**
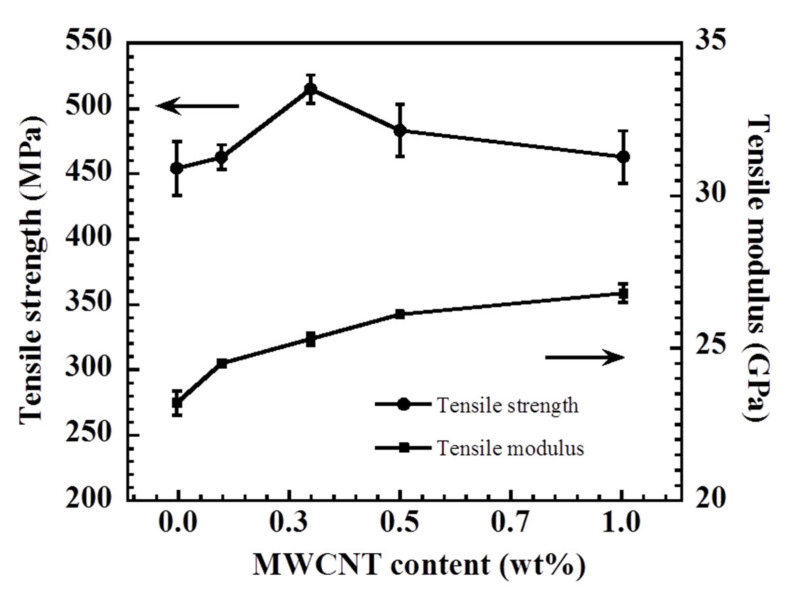
Summary of tensile property data collected for aramid fabrics that were reinforced with poly(BA-a/PU) composites containing different loadings of MWCNTs.

**Figure 6 polymers-13-02779-f006:**
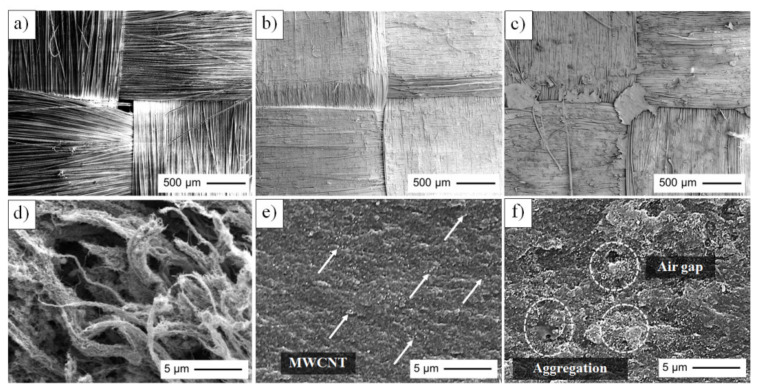
SEM micrographs recorded for (**a**) a pristine aramid fabric and poly(BA-a/PU) composites with (**b**) 0.3 wt% or (**c**) 0.5 wt% of MWCNTs. Magnification = 100×. (**d**) neat MWCNTs and a poly(BA-a/PU) composite with different MWCNT loadings: (**e**) 0.3 wt%, (**f**) 0.5 wt% of MWCNT. Magnification = 5000×.

**Figure 7 polymers-13-02779-f007:**
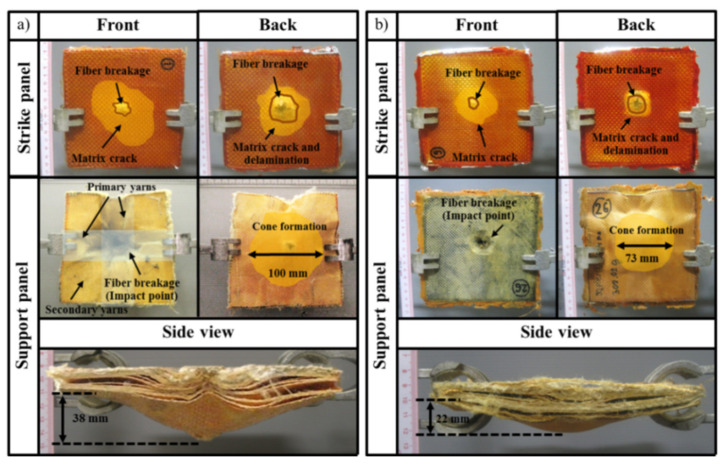
Ballistic-impact response of the specimens that consisted of a strike panel of S-glass fabric that was reinforced with poly(BA-a) and a support panel of an aramid fabric that was reinforced with a poly(BA-a/PU) composite (**a**) without MWCNTs or (**b**) containing 0.3 wt% of MWCNTs after being impacted by a 7.62 × 51 mm^2^ projectile.

**Figure 8 polymers-13-02779-f008:**
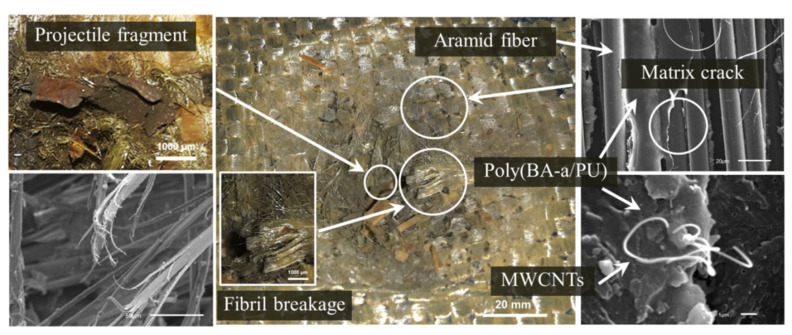
Ballistic-impact response of an aramid fabric-reinforced with a poly(BA-a/PU) composite containing 0.3 wt% of MWCNTs after being impacted.

**Figure 9 polymers-13-02779-f009:**
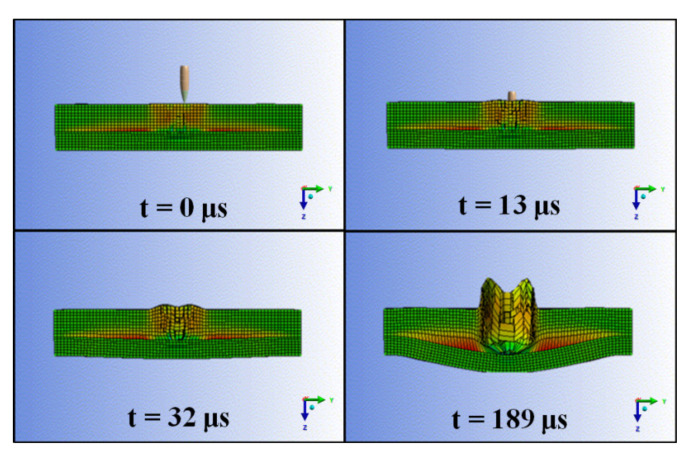
Numerical simulations results obtained for an impacted composite over time.

**Figure 10 polymers-13-02779-f010:**
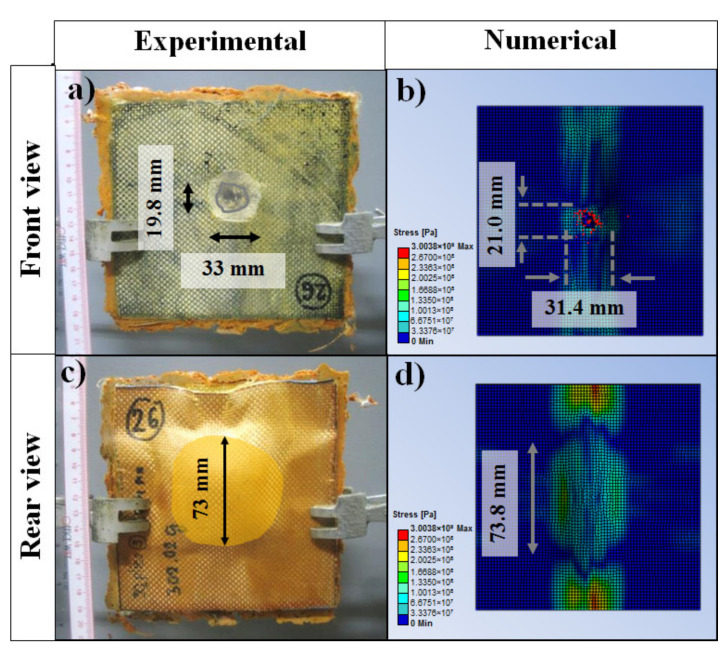
Comparison of damage areas as determined by (**a**,**c**) experiment or by (**b**,**d**) numerical simulation. See text for more details.

**Figure 11 polymers-13-02779-f011:**
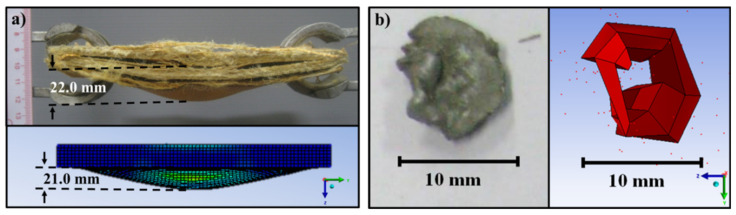
Comparison of (**a**) the depth of penetration and (**b**) the projectile shape after impact as determined by experiment or by numerical simulation. See text for more details.

**Figure 12 polymers-13-02779-f012:**
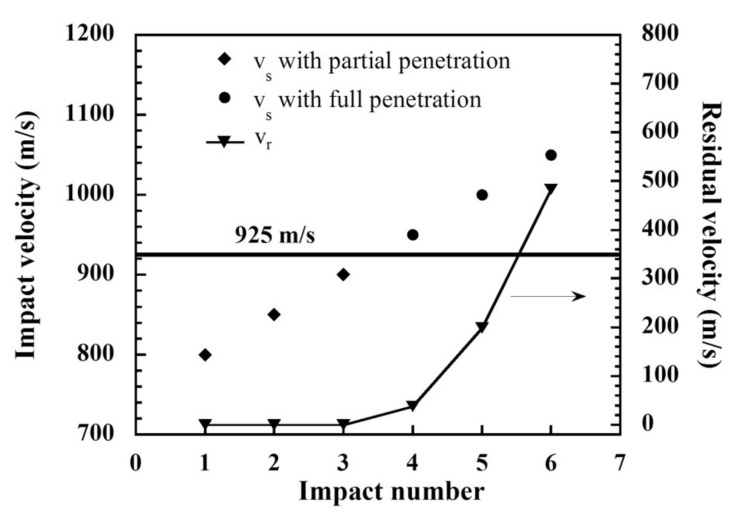
Plot of impact velocity versus impact number used to determine the ballistic limit velocity after being impacted by a 7.62 × 51 mm^2^ projectile.

**Table 1 polymers-13-02779-t001:** Summary of key properties for the material used in the 7.62 × 51 mm^2^ projectile.

Lead Core	Value	Properties	Value
Density	11,340 kg m^−1^	Derivative dG/dP	1
Shear modulus	8600 MPa	Derivative dG/dT	−9.976 MPa/°C
Plasticity	Steinberg Guinan Strength	Derivative dY/dP	0.0009304
Initial yield stress	8 MPa	Parameter C1	2006 m/s
Hardening constant	110	Parameter S1	1.429
Failure Maximum equivalent plastic strain	2	Parameter quadratic S2	0 s/m
Melting temperature	760 K	Gruneisen coefficient	2.74
**Brass Jacket**	**Value**	**Properties**	**Value**
Density	8450 kg m^−1^	Parameter C1	3726 m/s
Shear modulus	3000 MPa	Parameter S1	1.434
Gruneisen coefficient	2.74		

**Table 2 polymers-13-02779-t002:** Impact velocity, residual velocity, energy absorption and ballistic limit velocity values determined for aramid fabrics that were reinforced with poly(BA-a/PU) composites containing various quantities of MWCNTs.

MWCNT Content (wt%)	Sample Number	*V_s_* (m/s)	*V_r_* (m/s)	*E_Abs_* (J)	Average *E_Abs_* (J)	*V_bl_* (m/s)	Average *V_bl_* (m/s)
0.0	1	842.5	825.1	140.0	143.4 ± 22.8	170.3	171.8 ± 13.6
	2	840.2	825.6	117.3		156.0	
	3	851.4	830.1	172.8		189.3	
0.3	1	848.3	824.8	189.7	190.2 ± 35.5	198.3	198.5 ± 2.8
	2	859.2	835.1	197.0		202.1	
	3	854.1	831.5	183.8		195.2	

**Table 3 polymers-13-02779-t003:** A summary of material properties for the multilayered composites [44].

Properties		1st Layer (Strike Panel) ^a^	2nd Layer (Support Panel) ^b^
Thickness, mm		13.1	8.5
Density, g/cm^3^		2.2	1.6
Young modulus, kPa	$E_{11}$	7 × 10^7^	2.6 × 10^7^
	$E_{22}$	7 × 10^7^	2.6 × 10^7^
	$E_{33}$	7 × 10^6^	1.7 × 10^6^
Poisson’s ratio	$\nu_{12}$	0.12	0.07
	$\nu_{23}$	0.4	0.698
	$\nu_{31}$	0.2	0.075
Strength: Shear modulus, kPa	$G_{12}$	6 × 10^6^	4 × 10^5^
	$G_{23}$	6 × 10^6^	1.7 × 10^4^
	$G_{31}$	6 × 10^5^	1.7 × 10^4^
Failure: Tensile failure stress or strain	$\sigma_{11}\;or\;\varepsilon_{11}$	6 × 10^5^	0.07
	$\sigma_{22}\;or\;\varepsilon_{22}$	6 × 10^5^	0.07
	$\sigma_{33}\;or\;\varepsilon_{33}$	7 × 10^4^	0.02

^a^ 1st layer is S-glass fabric that was reinforced with a poly(BA-a) composite. ^b^ 2nd layer is an aramid fabric that was reinforced with a poly(BA-a/PU) composite containing MWCNTs.

**Table 4 polymers-13-02779-t004:** Comparison on thickness and areal weight density of our fiber reinforced poly(BA-a/PU) nanocomposite and other hard ballistic armors.

Hard Ballistic Armors	Thickness (mm)	Areal Weight Density (g/cm^2^)	Ref.
Fiber reinforced poly(BA-a/PU) nanocomposite	21.6	4.0	Present study
Rolled Homogenous Armour (RHA) steel	18.0	14.1	[45]
Ceramic/metal composite armor	25.0	5.0	[46]
Aluminium/steel alloy armor	12.0	10.0	[47]

## Data Availability

Data is contained within the article.

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
