# Peer review of "Impact Response of Aramid Fabric-Reinforced Polybenzoxazine/Urethane Composites Containing Multiwalled Carbon Nanotubes Used as Support Panel in Hard Armor"

_polymers, 2021, doi:10.3390/polym13162779_

Round 1
Reviewer 1 Report
The manuscript "Impact Response of Aramid Fabric Reinforced Polybenzoxazine/Urethane Composites Containing Multiwalled Carbon Nanotubes Using as Support Panel in Hard Armor" shows interesting results. There some minor question to address
If you show and describe signals in FTIR add the references
The author took max 1 wt% MWCNT. How are high loads affecting the composites? Are there any maximum where negative effects taking place?
Are the MWCNT aligned in the composites?
Figure 6. The scale bar is not visible in the SEM please add at least such in Figure capture
If you doing the ballistic test you shoot from one direction. What effect has it if you use other angles?
It would be beneficial to show some Tables of other works made (even those in market) and your work to have a comparison how well yours fitting in real application. Also add in such table the thickness and the weight of the armor.
Do your armor have stab-proof capability?
Reviewer 2 Report
Dear Authors,
I found your manuscript interesting, well prepared, describing properly designed experiments, correct discussion, and explained results.
However, some issues can be corrected (mainly editorial):
- line 118 - I think in case of area weight density of aramid fabric weave the unit should be g/m2 instead of g/cm2
- line 161 - gauge instead of gage?
- line 194 - specimens dimensions 150 x 150 should be in mm2 and not in mm3
- line 460 (fig. 12): I think something is wrong in legend: there is a rounded dot and diamond, whereas on the plot is a triangle and diamond
Regards!
